# In vivo experiments do not support the charge zipper model for Tat translocase assembly

**Felicity Alcock[†]\*, Merel PM Damen[†‡], Jesper Levring[†], Ben C Berks\***

Department of Biochemistry, University of Oxford, Oxford, United Kingdom

**Abstract** The twin-arginine translocase (Tat) transports folded proteins across the bacterial cytoplasmic membrane and the plant thylakoid membrane. The Tat translocation site is formed by substrate-triggered oligomerization of the protein TatA. Walther and co-workers have proposed a structural model for the TatA oligomer in which TatA monomers self-assemble using electrostatic 'charge zippers' (*Cell* (2013) **132:** 15945). This model was supported by in vitro analysis of the oligomeric state of TatA variants containing charge-inverting substitutions. Here we have used live cell assays of TatA assembly and function in *Escherichia coli* to re-assess the roles of the charged residues of TatA. Our results do not support the charge zipper model. Instead, we observe that substitutions of charged residues located in the TatA amphipathic helix lock TatA in an assembled state, suggesting that these charged residues play a critical role in the protein translocation step that follows TatA assembly.

DOI: https://doi.org/10.7554/eLife.30127.001

**\*For correspondence:** felicity.
alcock@bioch.ox.ac.uk (FA); ben.
berks@bioch.ox.ac.uk (BCB)

[†]These authors contributed
equally to this work

**Present address:** [‡]Amsterdam
Institute for Molecules,
Medicines and Systems, Vrije
Universiteit Amsterdam,
Amsterdam, The Netherlands

**Competing interests:** The
authors declare that no
competing interests exist.

**Reviewing editor:** Reid Gilmore,
University of Massachusetts
Medical School, United States

## Introduction

The Tat protein translocase is found in the bacterial cytoplasmic membrane, plant thylakoid membrane, and the inner membrane of some mitochondria (*Carrie et al., 2016*; *Celedon and Cline, 2013*; *Palmer and Berks, 2012*). Remarkably, Tat transports fully folded proteins across these tightly sealed membrane systems. The well-studied Tat system of *Escherichia coli* requires the three membrane proteins TatA, TatB and TatC (*Berks, 2015*; *Cline, 2015*). Binding of substrate proteins to a TatBC receptor complex triggers the proton-motive force (PMF)-dependent oligomerization of TatA protomers from a membrane pool onto the substrate-TatBC complex (*Alcock et al., 2013*; *Dabney-Smith et al., 2006*; *Rose et al., 2013*). The assembled TatA oligomer promotes translocation of the substrate protein across the membrane.

Elucidating the structure of the transiently formed TatA oligomer is key to understanding how folded proteins can cross a membrane bilayer. Although various models have been proposed (*Brüser and Sanders, 2003*; *Cline, 2015*; *Dabney-Smith et al., 2003*; *Gohlke et al., 2005*; *Greene et al., 2007*; *Palmer and Berks, 2012*; *Rodriguez et al., 2013*), the structure of the TatA oligomer remains enigmatic. A single TatA molecule comprises an N-terminal transmembrane helix (TMH), a basic amphipathic helix (APH) lying at the membrane surface, and a natively unstructured tail termed the densely charged region (DCR) (*Figure 1A,B*) (*Hu et al., 2010*; *Rodriguez et al., 2013*; *Walther et al., 2010*).

Walther and co-workers recently presented the concept of 'charge zippers' as a new structural principle to explain assembly of the TatA oligomer (*Walther et al., 2013*). This model arises from the observation that the APH and DCR of TatA molecules have complementary charge patterns (*Figure 1A*). Salt bridge formation between some of these oppositely charged residues was proposed to mediate folding and oligomerisation of TatA resulting in a transport pore as shown in *Figure 1C,D*.

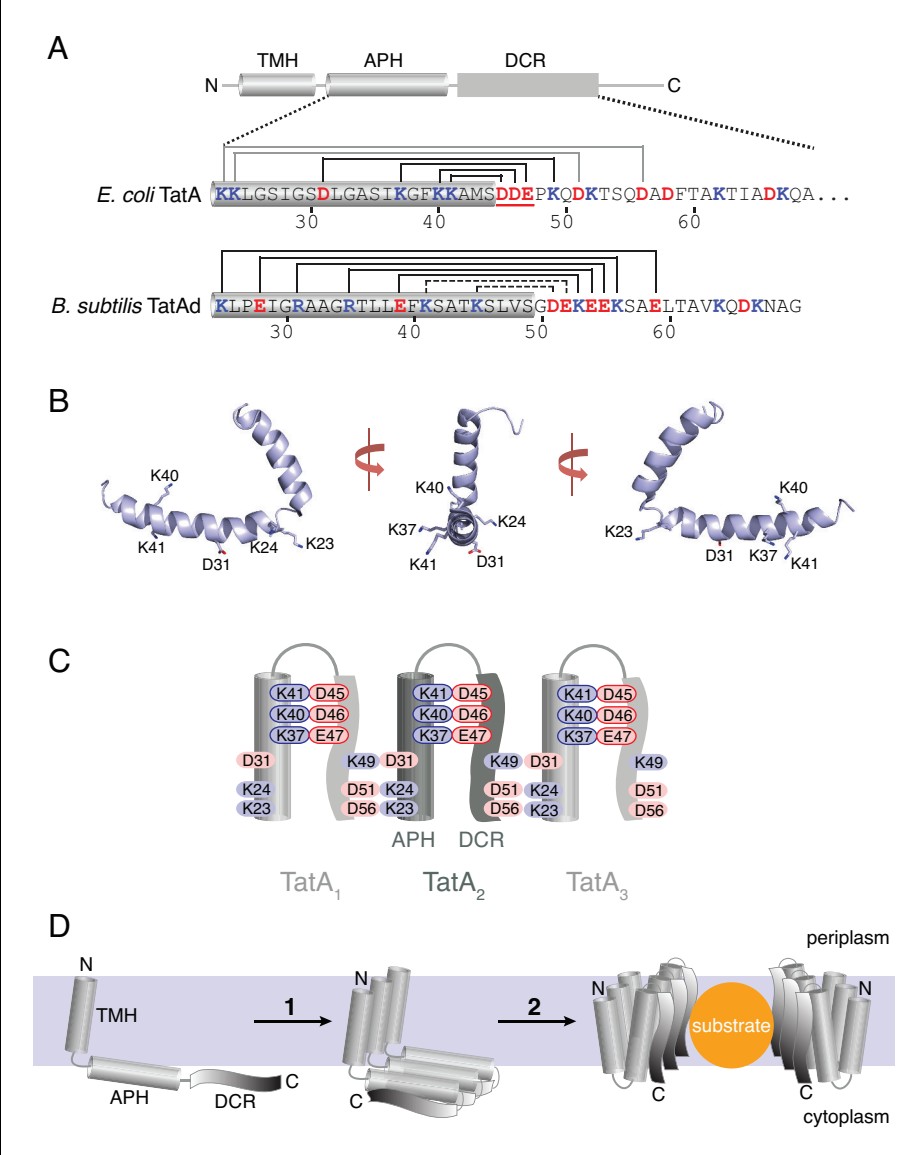

**Figure 1.** The charge zipper model for TatA assembly. (**A**) Cartoon showing the TatA domain structure, comprising a transmembrane helix (TMH), an amphipathic helix (APH), and a densely-charged region (DCR). Below the cartoon are shown the amino acid sequences for *E. coli* TatA (top) and *B. subtilis* TatAd (bottom) starting at the beginning of the APH and with acidic (red) and basic (blue) residues indicated. The APH is assigned in each case from the corresponding NMR structures (PDB 2MN7 and 2L16) and is indicated by a gray cylinder. The *E. coli* TatA sequence is C-terminally truncated to residue 69. Salt bridge pairs predicted by Walther and co-workers are indicated above each sequence. For *E. coli* TatA the predicted salt bridge pairs tested in this study are indicated in black and the acidic DDE motif targeted in this study is underlined. For *B. subtilis* TatAd, the assigned intra- and inter-molecular pairs are distinguished using dotted or solid lines respectively. (**B**) Structure of *E. coli* TatA (PDB 2MN7) shown in three orientations with the charged APH side chains indicated. (**C**) Schematic diagram of the charge zipper model for TatA folding and assembly applied to *E. coli* TatA. Folding back the DCR against the APH is proposed to allow pairing of amino acids with complementary charges to form either intra-molecular or inter-molecular salt bridges. Three adjacent TatA protomers are shown with the residues of the acidic DDE motif and their potential salt bridge partners outlined in red and blue respectively. The charge zipper model does not predict which residue pairs would form inter- and intra-molecular salt bridges and one of several possible configurations is represented here. (**D**) The salt bridges shown in (**C**) are proposed to mediate self-assembly of multiple adjacent TatA molecules (1). The assembled APH/DCR units are then proposed to insert across the membrane to form the passage for substrate protein transport (2).

DOI: https://doi.org/10.7554/eLife.30127.002

To experimentally test key predictions of the charge zipper model, Walther and co-workers disrupted proposed salt bridges of *Bacillus subtilis* TatAd. Single charge inversion substitutions were reported to impede TatA oligomerization. Crucially, for four of the seven predicted salt bridges, they reported that TatA assembly could be restored by simultaneously inverting the charge on the predicted partner residue, thereby re-establishing a salt bridge. These observations appear to provide strong support for the charge zipper model. However, these experiments were carried out in vitro, with the degree of TatA oligomerization assessed by blue native-PAGE of detergent-solubilized membranes. Whether TatA complexes observed in detergent solution can be directly related to physiological TatA oligomers remains a matter of debate since TatA complexes can be observed in detergent extracts in the absence of substrate proteins, or the TatBC complex, or the PMF, all of which are required for formation of native TatA oligomers (*Gohlke et al., 2005*; *Palmer and Berks, 2012*). In addition, the oligomeric states of TatA observed in detergent solution are known to be influenced by the detergent to protein ratio (*Cline and Mori, 2001*; *Rodriguez et al., 2013*) casting doubt on whether detergent extraction accurately captures the TatA oligomeric state present in the membrane before solubilization. In order to address these concerns we have revisited Walther and co-workers' tests of the charge zipper model but in a native physiological context.

## Results

### Analysis of Tat transport activity in strains lacking one or more charge zipper pairs

The predicted salt bridge pairs of *E. coli* TatA are shown in *Figure 1A* (*Walther et al., 2013*). We examined the four potential salt bridge pairs where both partners are in the functionally essential first 50 residues of TatA (*Lee et al., 2002*; *Warren et al., 2009*). Because amino acid substitutions which prevent translocase assembly should also block transport we examined the Tat transport activity of variants at these target salt bridges.

Transport activities were assessed using three established assays. First, we overproduced the *E. coli* Tat substrate CueO and assessed how much of the protein reached the periplasm. Second, we measured growth rates with the anaerobic electron acceptor trimethylamine N-oxide (TMAO), which is metabolized by Tat-dependent periplasmic enzymes. Third, we analysed cell growth on SDS-containing medium because Tat null strains have a defective cell envelope that renders them sensitive to SDS (*Stanley et al., 2001*).

We first examined the proposed salt bridge between residues D31 and K49. The expectation from the charge zipper model is that removing either end of a proposed salt bridge would result in identical Tat transport phenotypes. However, while a D31K variant was unable to export CueO or grow on TMAO, as anticipated if D31 is part of a zipper bridge, a K49D substitution still permitted significant transport of CueO and close to wild-type growth on TMAO (*Figure 2A,B*). The previous in vitro tests of the charge zipper model (*Walther et al., 2013*) led us to expect that simultaneous charge reversal at both ends of the salt bridge will restore salt bridge function. However, combining the K49D substitution with the inactivating D31K substitution failed to restore export of CueO or growth on TMAO (*Figure 2A,B*). When these various salt bridge variants were assessed using the SDS growth assay, which can detect very low levels of Tat activity, transport was now observed for D31K-substituted Tat, although at lower levels than with the wild-type protein. Nevertheless, the D31K/K49D variant, rather than restoring wild-type Tat transport activity to D31K as predicted by the charge zipper model, exhibited even poorer growth (*Figure 2C*). To account for the possibility that the salt bridge partner of D31 had been misassigned, we also considered K52 as a potential partner. However, we found that, as with K49D, a K52D substitution had only a moderate effect on Tat transport and did not restore activity to the D31K variant (*Figure 2A,B*). Taken together these experiments provide no evidence that the decreased activity of TatA D31K results from disrupting a salt bridge.

We next examined the consequence of simultaneously removing the three potential salt bridges formed by the acidic $D_{45}D_{46}E_{47}$ motif (*Figure 1A,C*), which is already implicated in Tat function (*Fröbel et al., 2011*; *Greene et al., 2007*; *Hicks et al., 2005*; *Warren et al., 2009*). Disrupting these three salt bridge pairs on the APH side (EDD variant) or the DCR side (KKK variant) abolished activity in our CueO export and TMAO growth assays (*Figure 2A,B*). Nevertheless, a TatA EDD/KKK variant,

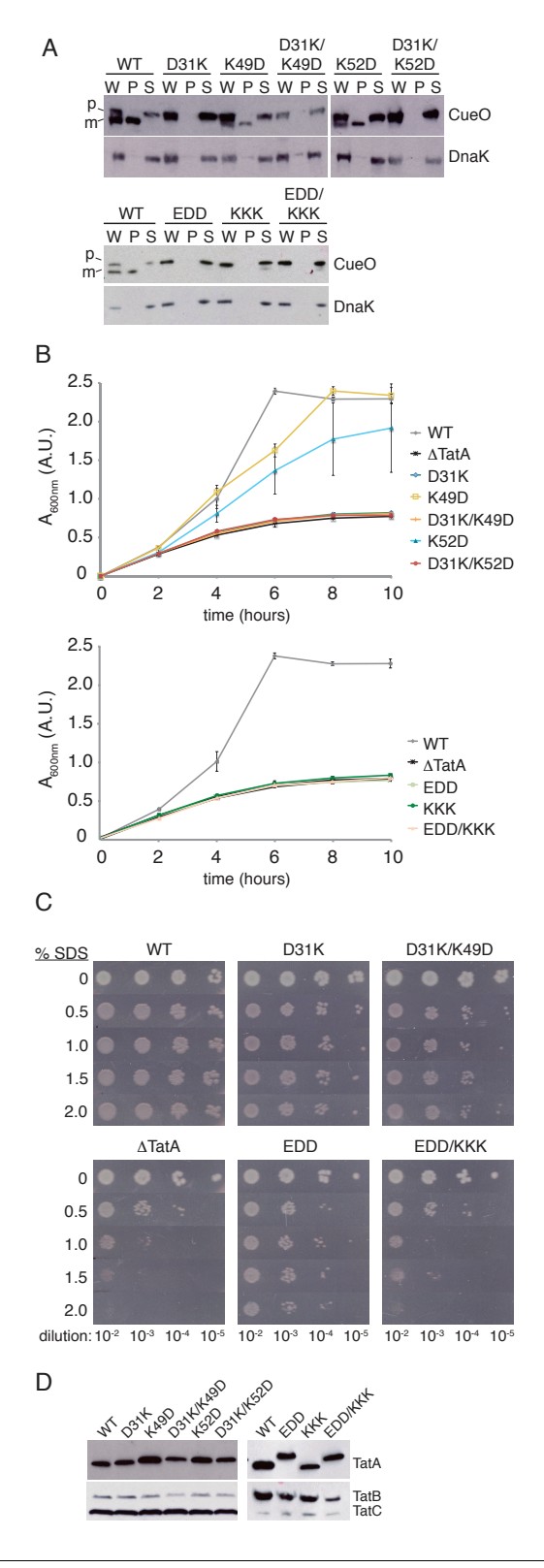

**Figure 2.** Transport activity of TatA charge zipper variants. All TatA variants were expressed from the phage lambda attachment site in the ΔtatA ΔtatE strain JARV16. Mutant strains are identified by the amino acid substitutions present in TatA where EDD is K37E/K40D/K41D, and KKK is D45K/D46K/E47K. WT refers to the ΔtatE strain J1M1, and ΔTatA refers to the parental strain JARV16. (**A**) Whole cell (**W**), periplasm (**P**), and spheroplast (**S**)
*Figure 2 continued on next page*

*Figure 2 continued*

fractions of cells overproducing CueO from plasmid pQE80-CueO were subject to immunoblotting with antibodies against CueO or the cytoplasmic marker protein DnaK. m is the transported form of CueO from which the signal peptide has been removed and p the precursor protein. (B) Growth of the strains when cultured in LB/glycerol/TMAO medium under anoxic conditions. Error bars represent the S.E.M of three biological replicates. (C) Serial ten-fold dilutions of log-phase cultures were spotted onto LB-agar containing the indicated amount of SDS. (D) Immunoblot of membranes isolated from the strains used in A-C, probed with TatA antibodies (upper panels) or antibodies against TatB and TatC (lower panels).

DOI: https://doi.org/10.7554/eLife.30127.003

which inverts the potential salt bridges, failed to restore detectable activity. Using the more sensitive SDS assay, TatA EDD exhibited a low level of Tat transport activity that was clearly greater than the negative control (*Figure 2C*). However, restoration of the predicted charge pairs in TatA EDD/KKK completely abolished Tat transport instead of recovering activity. Thus, in contrast to the earlier in vitro analysis of the charge zipper model, restoring putative salt bridges by complementary charge reversal in the DCR not only fails to bring back transport activity to the APH variants, but even further reduces their activity.

Immunoblotting confirmed that each variant analysed was present in the membrane at a comparable level to the wild-type TatA protein (*Figure 2D*).

## Direct observation of TatA assembly in charge zipper mutants

We used live cell imaging of a previously described TatA-YFP fusion to directly test the ability of the TatA charge zipper variants to undergo substrate-induced oligomerization (*Alcock et al., 2013*). In resting cells the TatA-YFP fusion is present in the dispersed state, visualised as a halo of fluorescence (*Figure 3*). Overproduction of the Tat substrate CueO drives assembly of TatA-YFP oligomers which appear as bright, mobile fluorescent spots. If the proton motive force (PMF) is abolished by addition of the protonophore carbonyl cyanide *m*-chlorophenyl hydrazone (CCCP), the TatA oligomers dissociate and fluorescence reverts to the peripheral halo.

We found that all our charge zipper variants were able to assemble into fluorescent foci (*Figure 3*). While those charge zipper substitutions that retained significant Tat activity (K49D and K52D) fully replicated the TatA-YFP oligomerization behaviour of the wild-type protein, those charge zipper variants which had very low transport activity (D31K and KKK) remained in the assembled state in all conditions tested. Thus, whilst TatA oligomerization is possible in the low activity variants the assembly/disassembly cycle is perturbed.

As our experimental strains retain the TatA paralog TatE we considered the possibility that TatE, although present at very low levels, might be aiding assembly of the TatA zipper variants. Consequently, we repeated the imaging using strains lacking TatE. Under these conditions it is well-established that the wild type TatA-YFP protein exhibits perturbed behaviour, being visualised in the assembled state even in the absence of overproduced substrate or the presence of CCCP (*Alcock et al., 2013*; *Leake et al., 2008*). We found that both high and low activity charge zipper variants were still able to form TatA-YFP oligomers in the absence of TatE (*Figure 3*), which eliminates the possibility that TatE was mediating the TatA assembly observed for these variants. Importantly, oligomer formation by these TatA-YFP variants was in all cases strictly dependent on the presence of TatBC (*Figure 3*) showing that the charge zipper variants still assemble TatA via the physiological route, most likely triggered by endogenous substrates (*Alcock et al., 2013*), rather than through self-aggregation.

Although problems with expression level and proteolytic release of YFP meant the TatA-YFP EDD variant could not be directly compared to the other charge zipper variants, it also appears to form TatBC-dependent TatA-YFP assemblies in both the presence and absence of TatE (*Figure 3—figure supplement 1*).

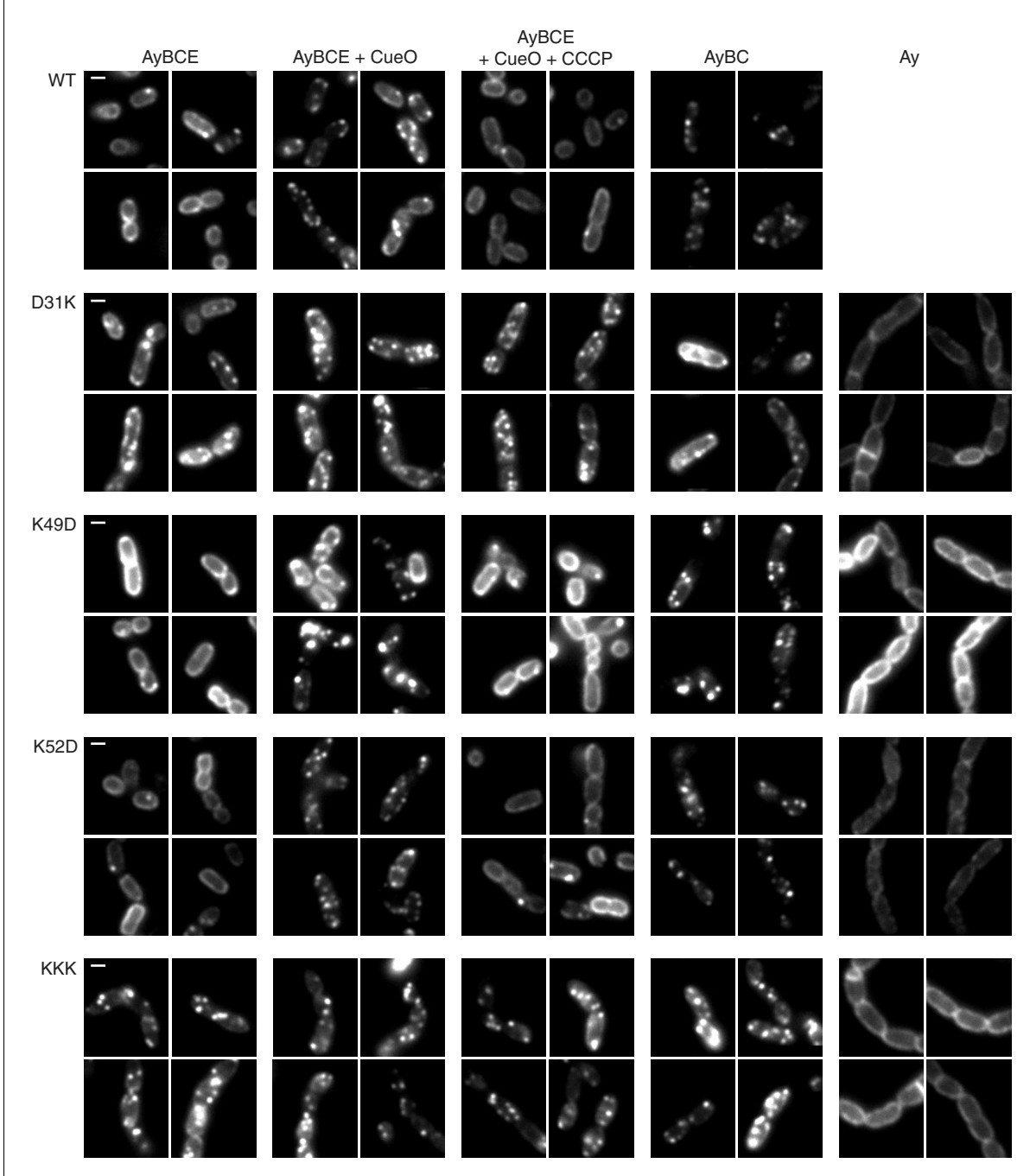

**Figure 3.** TatA oligomerization behavior of charge zipper variants. Representative fluorescence images of TatA-YFP in *E. coli* cells. A *tatA-yfp* fusion was expressed from the chromosome in three different backgrounds: ELV16 λAry which contains all other *tat* genes (designated AyBCE), JARV16 λAry which lacks *tatE* (designated AyBC), or DADE λAry which possesses no other *tat* genes (designated Ay). The TatA-YFP variant produced is indicated to the left of the panels, where WT is the parental protein and KKK is a D45K/D46K/E47K variant. Where indicated, CueO was overproduced from plasmid pQE80-CueO by adding IPTG to early exponential phase cultures for 30 min prior to imaging (+CueO columns). 50 μM CCCP was subsequently added, as indicated (+CCCP column), and the cells incubated for 30–45 min prior to imaging. Scale bar = 1 μm.

DOI: https://doi.org/10.7554/eLife.30127.004

The following figure supplement is available for figure 3:

**Figure supplement 1.** TatA oligomerization behavior of additional charge zipper variants.

DOI: https://doi.org/10.7554/eLife.30127.005

## Discussion

We have used physiologically relevant assays of TatA oligomerization and function to reassess Walther and co-workers' in vitro tests of the charge zipper model of TatA assembly (*Walther et al., 2013*). Our in-cell results fail to substantiate key outcomes of Walther and co-workers' tests.

Firstly, and most fundamentally, Walther and co-workers report that charge zipper substitutions affect TatA oligomerization. We found that none of our charge zipper substitutions prevented TatA oligomerization in live cells, even though the majority of our variants were confirmed to have severe functional defects. Secondly, Walther and co-workers report that TatA assembly defects caused by charge reversal substitutions at one end of a salt bridge can be repaired by reforming the salt bridge through concurrent charge reversal at the other end of the bridge. However we were unable to replicate these observations when assessing Tat transport activity. Finally, ablation of either end of a predicted charge zipper salt bridge should give rise to the same structural and functional defects. Yet for some of the putative salt bridge variants tested (D31K versus K49D or K52D) we see large differences in transport and oligomerization behavior between substitutions at each end of the salt bridge. Our variants target almost all of the charged residues in the APH and early DCR, so it is highly implausible that we have failed to affect the intermolecular charge zippers if these exist.

Our in vivo observations cast considerable doubt on the charge zipper model as a plausible mechanism for the assembly of TatA or other membrane protein homo-oligomers. Nevertheless, our data do show that the charged residues in the TatA APH and proximal DCR are critical for Tat transport. Substitutions of these amino acids were observed to perturb the Tat transport cycle resulting in TatA oligomers that are resistant to loss of the PMF, and that form even without substrate overproduction. The same behavior has previously been reported following substitution of the highly conserved and functionally essential APH residue Phe39 (*Alcock et al., 2013*; *Hicks et al., 2003*; *Hicks et al., 2005*). Thus, several substitutions in the APH lead to the accumulation of TatA in a stable oligomeric state. In each case the observed TatA oligomerization is not spontaneous, but triggered by endogenous Tat substrates, as demonstrated by the dependence on TatBC (*Figure 3*, (*Alcock et al., 2013*)). The subsequent accumulation of stably assembled TatA oligomers is most parsimoniously explained by suggesting that the APH is required for a step in the transport cycle between TatA oligomerization and TatA disassembly, and that this step is disrupted in the APH variants. Logically, this mechanistic step is likely to be the physical movement of the substrate protein across the membrane bilayer.

## Materials and methods

**Key resources table**

| Reagent type (species) or resource | Designation | Source or reference | Identifiers | Additional information |
|---|---|---|---|---|
| strain, strain background (*Escherichia coli*) | MC4100 | *Casadaban and Cohen, 1979* | | |
| strain, strain background (*E. coli*) | MC1061 | *Casadaban and Cohen, 1980* | | |
| strain, strain background (*E. coli*) | J1M1 | *Sargent et al., 1998* | | MC4100 Δ*tatE* |
| strain, strain background (*E. coli*) | JARV16 | *Sargent et al., 1999* | | MC4100 Δ*tatA* Δ*tatE* |
| strain, strain background (*E. coli*) | ELV16 | *Sargent et al., 1999* | | MC4100 Δ*tatA* |
| strain, strain background (*E. coli*) | DADE | *Wexler et al., 2000* | | MC4100 Δ*tatABCD* Δ*tatE* |

| strain, strain background (*E. coli*) | MΔABC | Alcock et al., 2013 | MC4100 Δ*tatABC::apra* |
|---|---|---|---|
| strain, strain background (*E. coli*) | JARV16 λA D31K | This paper | JARV16 *attB*::P$_{tatA}$*tatA*$^{D31K}$ (kan$^r$) |
| strain, strain background (*E. coli*) | JARV16 λA K49D | This paper | JARV16 *attB*::P$_{tatA}$*tatA*$^{K49D}$ (kan$^r$) |
| strain, strain background (*E. coli*) | JARV16 λA D31K/K49D | This paper | JARV16 *attB*::P$_{tatA}$*tatA*$^{D31K,K49D}$ (kan$^r$) |
| strain, strain background (*E. coli*) | JARV16 λA K52D | This paper | JARV16 *attB*::P$_{tatA}$*tatA*$^{K52D}$ (kan$^r$) |
| strain, strain background (*E. coli*) | JARV16 λA D31K/K52D | This paper | JARV16 *attB*::P$_{tatA}$*tatA*$^{D31K,K52D}$ (kan$^r$) |
| strain, strain background (*E. coli*) | JARV16 λA EDD | This paper | JARV16 *attB*::P$_{tatA}$*tatA*$^{K37E,K40D,K41D}$ (kan$^r$) |
| strain, strain background (*E. coli*) | JARV16 λA KKK | This paper | JARV16 *attB*::P$_{tatA}$*tatA*$^{D45K,D46K,E47K}$ (kan$^r$) |
| strain, strain background (*E. coli*) | JARV16 λA EDD/KKK | This paper | JARV16 *attB*::P$_{tatA}$*tatA*$^{K37E,K40D,K41D,D45K,D46K,E47K}$ (kan$^r$) |
| strain, strain background (*E. coli*) | JARV16 λAry | This paper | JARV16 *attB*::P$_{tatA}$*tatA*$^{-}$EAK-*eyfp*$^{A206K}$ (kan$^r$) |
| strain, strain background (*E. coli*) | JARV16 λAry D31K | This paper | JARV16 *attB*::P$_{tatA}$*tatA*$^{D31K-}$EAK-*eyfp*$^{A206K}$ (kan$^r$) |
| strain, strain background (*E. coli*) | JARV16 λAry K49D | This paper | JARV16 *attB*::P$_{tatA}$*tatA*$^{K49D-}$EAK-*eyfp*$^{A206K}$ (kan$^r$) |
| strain, strain background (*E. coli*) | JARV16 λAry K52D | This paper | JARV16 *attB*::P$_{tatA}$*tatA*$^{K52D-}$EAK-*eyfp*$^{A206K}$ (kan$^r$) |
| strain, strain background (*E. coli*) | JARV16 λAry EDD | This paper | JARV16 *attB*::P$_{tatA}$*tatA*$^{K37E,K40D,K41D-}$EAK-*eyfp*$^{A206K}$ (kan$^r$) |
| strain, strain background (*E. coli*) | JARV16 λAry KKK | This paper | JARV16 *attB*::P$_{tatA}$*tatA*$^{D45K,D46K,E47K-}$EAK-*eyfp*$^{A206K}$ (kan$^r$) |
| strain, strain background (*E. coli*) | ELV16 λAry D31K | This paper | ELV16 *attB*::P$_{tatA}$*tatA*$^{D31K-}$EAK-*eyfp*$^{A206K}$ (kan$^r$) |
| strain, strain background (*E. coli*) | ELV16 λAry K49D | This paper | ELV16 *attB*::P$_{tatA}$*tatA*$^{K49D-}$EAK-*eyfp*$^{A206K}$ (kan$^r$) |
| strain, strain background (*E. coli*) | ELV16 λAry K52D | This paper | ELV16 *attB*::P$_{tatA}$*tatA*$^{K52D-}$EAK-*eyfp*$^{A206K}$ (kan$^r$) |
| strain, strain background (*E. coli*) | ELV16 λAry EDD | This paper | ELV16 *attB*::P$_{tatA}$*tatA*$^{K37E,K40D,K41D-}$EAK-*eyfp*$^{A206K}$ (kan$^r$) |
| strain, strain background (*E. coli*) | ELV16 λAry KKK | This paper | ELV16 *attB*::P$_{tatA}$*tatA*$^{D45K,D46K,E47K-}$EAK-*eyfp*$^{A206K}$ (kan$^r$) |

| strain, strain background (*E. coli*) | DADE λAry D31K | This paper | | DADE *attB*::P$_{tatA}$*tatA*$^{D31K}$-*EAK-eyfp*$^{A206K}$ (kan$^r$) |
|---|---|---|---|---|
| strain, strain background (*E. coli*) | DADE λAry K49D | This paper | | DADE *attB*::P$_{tatA}$*tatA*$^{K49D-}$*EAK-eyfp*$^{A206K}$ (kan$^r$) |
| strain, strain background (*E. coli*) | DADE λAry K52D | This paper | | DADE *attB*::P$_{tatA}$*tatA*$^{K52D-}$*EAK-eyfp*$^{A206K}$ (kan$^r$) |
| strain, strain background (*E. coli*) | DADE λAry KKK | This paper | | DADE *attB*::P$_{tatA}$*tatA* $^{D45K,D46K,E47K-}$*EAK-eyfp*$^{A206K}$ (kan$^r$) |
| strain, strain background (*E. coli*) | MΔABC λAry EDD | This paper | | MC4100 Δ*tatABC*::*apra attB*::P$_{tatA}$*tatA* $^{K37E,K40D,K41D-}$*EAK-eyfp*$^{A206K}$ (kan$^r$) |
| antibody | anti-CueO | This paper | | Rabbit poly clonal against CueO mature domain. Affinity purified (1:200) |
| antibody | anti-DnaK | Abcam | Abcam ab69617; Clone 8E2/2 | Mouse monoclonal (1:20000) |
| antibody | anti-TatA | *Alcock et al., 2016* | | Rabbit polyclonal against TatA soluble domain (1:5000) |
| antibody | anti-TatB | *Alcock et al., 2016* | | Rabbit polyclonal against TatB C-terminal peptide. Affinity purified (1:400) |
| antibody | anti-TatC | *Alcock et al., 2016* | | Rabbit polyclonal against TatC C-terminal peptide. Affinity purified (1:1000) |
| recombinant DNA reagent | pQE80-CueO | *Leake et al., 2008* | | Synthesis of *E. coli* CueO with a C-terminal his$_6$ tag |
| recombinant DNA reagent | pKSUniA | *Koch et al., 2012* | | pBluescript-based vector carrying P$_{tatA}$-*tatA* |
| recombinant DNA reagent | pBSTatAry | *Alcock et al., 2013* | | pBluescript-based vector carrying P$_{tatA}$-*tatA-EAK-eyfp*$^{A206K}$ |
| recombinant DNA reagent | pRS552 | *Simons et al., 1987* | | Shuttle vector for integration of DNA at the *E. coli* phage lambda attachment site (*attB*) |
| recombinant DNA reagent | λRS45 | *Simons et al., 1987* | | Phage for integration of DNA at the *E. coli* phage lambda attachment site (*attB*) |
| sequence-based reagent | TatA D31K F | Sigma-Aldrich, St. Louis, Missouri | | Oligonucleotide GGCTCCATCGGTTC CAAACTTGGTGCGTCGATC |
| sequence-based reagent | TatA D31K R | Sigma-Aldrich | | Oligonucleotide GATCGACGCACCA AGTTTGGAACCGATGGAGCC |
| sequence-based reagent | TatA K49D F | Sigma-Aldrich | | Oligonucleotide CAATGAGCGATGATGAACC AGATCAGGATAAAACCAGTCAGG |
| sequence-based reagent | TatA K49D R | Sigma-Aldrich | | Oligonucleotide CCTGACTGGTTTTATCCT GATCTGGTTCATCATCGCTCATTG |
| sequence-based reagent | TatA K52D F | Sigma-Aldrich | | Oligonucleotide GAACCAAAGCAGGA TGATACCAGTCAGGATGCTG |
| sequence-based reagent | TatA K52D R | Sigma-Aldrich | | Oligonucleotide CAGCATCCTGACT GGTATCATCCTGCTTTGGTTC |
| sequence-based reagent | TatA EDD F | Sigma-Aldrich | | Oligonucleotide CTTGGTGCGTCGATCGAAGG CTTTGATGATGCAATGAGCGATGATG |
| sequence-based reagent | TatA EDD R | Sigma-Aldrich | | Oligonucleotide CATCATCGCTCATTGCATC ATCAAAGCCTTCGATCGACGCACCAAG |
| sequence-based reagent | TatA KKK F | Sigma-Aldrich | | Oligonucleotide GCTTTAAAAAAGCAATGAG CAAAAAGAAACCAAAGCAGGATAAAACC |
| sequence-based reagent | TatA KKK R | Sigma-Aldrich | | Oligonucleotide GGTTTTATCCTGCTTTGG TTTCTTTTTGCTCATTGCTTTTTTAAAGC |

| sequence-based reagent | TatA zip2 F | Sigma-Aldrich | Oligonucleotide CTTGGTGCGTCGATCGAAGGCT TTGATGATGCAATGAGCAAAAAG |
| sequence-based reagent | TatA zip2 R | Sigma-Aldrich | Oligonucleotide CTTTTTGCTCATTGCATCATCAA AGCCTTCGATCGACGCACCAAG |

## Strains and plasmids

All strains, plasmids, oligonucleotides, and antibodies used in this study are listed in the Key Resources Table. Codon changes were carried out using the Quikchange method (Agilent, Santa Clara, California) with plasmid pKSUniA (for *tatA* mutants) or plasmid pBSTatAry (for *tatA-yfp* mutants) as the template. The mutated alleles were moved into plasmid pRS552 by restriction cloning with BamHI and EcoRI. An overnight culture of MC1061 cells transformed with the pRS552 construct of interest was mixed with phage λRS45 and the resulting phage lysate used to infect the MC4100-derivative background strain. Colonies were assessed for the presence of the desired gene insert by screening for kanamycin-resistance and for the absence of pRS552 by ampicillin-sensitivity. The constructed strains were further validated by amplification of the *tat* and *attB* loci, and by sequencing the PCR product amplified from the *attB* locus.

Unless otherwise indicated, cells were cultured in LB medium (*Sambrook and Russell, 2001*) at 37°C in a shaking incubator with the following antibiotic concentrations: ampicillin (100 µg /ml), kanamycin (50 µg/ml).

## CueO export assay

Overnight cultures of cells freshly transformed with plasmid pQE80-CueO were diluted 1:40 into fresh medium, then incubated for 1h30 at 37°C, with 1 mM IPTG added after 45 min of growth. Cells were harvested by centrifugation and resuspended in 10 mM Tris.Cl, 150 mM NaCl, pH 7.3 with cell densities normalized according to $OD_{600nm}$. Equal volumes of the cell suspensions were then centrifuged, and the cell pellets resuspended in 400 µl SET buffer (17.12% sucrose (w/v), 3 mM EDTA, 10 mM Tris.Cl, pH 7.3). 133 µl lysozyme (3 mg/ml in water) and 400 µl ice-cold water were immediately added. Samples were incubated for 20 min at 37°C, and spheroplasts were separated from the released periplasm by centrifugation. Samples were analyzed by immunoblotting for CueO and DnaK. The data presented are representative of at least three independent experiments.

## Anoxic growth on TMAO

Three biological replicates of each strain were grown overnight aerobically in LB. 250 µl of each pre-culture was used to inoculate 29 ml LB containing 0.4% TMAO (w/v) and 0.5% glycerol (v/v) in a 30 ml universal tube. Cultures were incubated at 37°C without shaking.

## SDS sensitivity assay

An overnight pre-culture of each strain was diluted 1:37.5 into fresh LB and incubated for 1–2 hr at 37°C. The $OD_{600nm}$ of each culture was normalised to 0.125 and serial ten-fold dilutions were then spotted on to a series of LB-agar plates containing different concentrations of SDS. The plates were incubated overnight at 37°C. The data presented are representative of at least three independent experiments.

## Isolation of membranes for immunoblot analysis

Logarithmic phase cell cultures were centrifuged, and cells were suspended in buffer A (50 mM Tris.Cl, 150 mM NaCl, pH 7.4) containing 0.2 mg/ml lysozyme. Cells were lysed by sonication, cell debris was removed by centrifugation (10 min at 3750 x g) and membranes were isolated by ultracentrifugation (1 hr at 100,000 x g). Membranes were resuspended in buffer A, and their concentrations were measured and normalized using the DC assay (Bio-Rad, Hercules, California).

## Fluorescence microscopy

Cells for microscopy were grown in LB, washed in M9/glucose medium (*Sambrook and Russell, 2001*), and immobilised on a tunnel slide using poly-L-lysine as previously described (*Alcock et al., 2013*). Cells were imaged using an inverted fluorescence microscope in HiLo configuration by excitation with a 532 nm laser. Emitted fluorescence was filtered through a dichroic and a 550 nm LP

emission filter as described previously (*Alcock et al., 2013*; *Huang et al., 2017*). Image stacks averaged over 40 ms were scaled to display 1500 arbitrary units (a.u.) as the minimum (black) and 7500 a.u. as the maximum (white), unless otherwise noted, and were exported as PNG files. Cell imaging panels show exemplar data from at least three independent cultures examined on different days.

## Acknowledgements

We thank Mark I Wallace and Hajra Basit for access to, and technical assistance with, fluorescence microscopy, Shee Chien Yong for assistance producing antibodies, and Tracy Palmer for comments on the manuscript.

## Additional information

### Funding

| Funder | Grant reference number | Author |
| --- | --- | --- |
| Wellcome | Investigator Award 107929/Z/15/Z | Ben C Berks |
| Biotechnology and Biological Sciences Research Council | BB/L002531/1 | Ben C Berks |
| European Commission | Erasmus Trainee Scheme | Merel PM Damen |
| Microbiology Society | Harry Smith Vacation Studentship VS15/10 | Jesper Levring |

The funders had no role in study design, data collection and interpretation, or the decision to submit the work for publication.

### Author contributions

Felicity Alcock, Conceptualization, Supervision, Investigation, Writing—original draft, Project administration, Writing—review and editing; Merel PM Damen, Jesper Levring, Funding acquisition, Investigation, Writing—review and editing; Ben C Berks, Conceptualization, Supervision, Funding acquisition, Writing—original draft, Project administration, Writing—review and editing

### Author ORCIDs

Felicity Alcock, https://orcid.org/0000-0002-3983-6097
Merel PM Damen, http://orcid.org/0000-0003-4233-2671
Ben C Berks, http://orcid.org/0000-0001-9685-4067

### Decision letter and Author response

Decision letter https://doi.org/10.7554/eLife.30127.007
Author response https://doi.org/10.7554/eLife.30127.008

## Additional files

### Supplementary files

• Transparent reporting form
DOI: https://doi.org/10.7554/eLife.30127.006

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
