## [Decision Letter]

Thank you for submitting your article "in vivo experiments do not support the charge zipper model for Tat translocase assembly" for consideration by *eLife*. Your article has been favorably evaluated by Richard Aldrich (Senior Editor) and three reviewers, one of whom, Reid Gilmore, is a member of our Board of Reviewing Editors. The reviewers have opted to remain anonymous.

The reviewers have discussed the reviews with one another and the Reviewing Editor has drafted this decision to help you prepare a revised submission.

Summary:

This manuscript from the Berks lab is a brief report describing in vivo tests of the charge zipper model for TatA oligomerization. The charge zipper model proposes that salt bridges formed between the APH and DCR segments of TatA are involved in formation of TatA oligomers which serve as the transport pore for Tat substrates. The Tat system goes through a cycle of activity that includes substrate binding to the BC complex, recruitment of TatA to BC and TatA oligomerization, then transport of the substrate and disassembly. Each step in the process has been extensively studied, yet no consensus for a mechanism has emerged. Berks and colleagues were the first to observe that the TatA APH segment was particularly sensitive to substitution mutations. Walther et al. conducted experiments that suggested that APH mutations reflect the requirement for salt bridges between APH and the DCR segment for assembly of TatA into oligomers. In some sense this hypothesis drew upon a flawed study by Gohlke et al. that TatA existed as a population of pore shaped oligomers that could be responsible for the transport of substrate with varying diameter. Even at the time of the Walther report, the detergent extraction methods employed were suspect and likely reported non-physiological states of TatA. In the intervening years better methods, both in vitro and in vivo, have reported on the blocks in the TatABC cycle and the assembly status of TatA. To their credit, the authors' 2013 paper (Rodrigues et al. 2013) pretty much laid to rest the notion that detergent solubilized TatA reflected the in situ form of TatA. Here Alcock et al. report sophisticated in vivo methods to determine if specific charged mutations in the APH and DCR affect TatA assembly and Tat substrate transport. The methodology is first rate, the conclusions straightforward and clear. The manuscript addresses an important controversy in the Tat transport field. As the 'charge zipper' model has been recently expanded to apply to other systems, an investigation of this model is timely. The reviewers concluded that a revised version of this manuscript would be appropriate for publication in *eLife*.

Essential revisions:

1) Sometimes these "refutation" papers don't get the most important point across. For this work, I think that the most important point is that the APH plays a crucial role in the translocation step. It further substantiates the point proposed in the earlier Alcock et al. 2013 paper that mutations in the APH causes an arrest at the assembled step and that the assembled state is stable even without the PMF, unless the substrate is transported.

2) The authors did a good job in the Alcock et al. 2013 paper explaining why assembly can be seen in an uninduced system (i.e. no overexpressed substrate) if the transport step is impaired. That argument is important in the present manuscript and should at least be referred to in the Discussion. The authors don't want to leave the impression that TatA assembles without substrate. Triggering of assembly by substrate binding is generally accepted in the field.

3) Figure 2. The spheroplast fraction of several mutants (TatA K49D and TatA K52D) has a band that comigrates with processed CueO. Is this band an aborted transport product? This band is not present or very faint in samples from the most defective mutants (TatA EDD/KKK) so it doesn't seem to be a cross-reactive protein or a cytosolic CueO degradation product. The authors should comment on this band particularly if it provides some insight into the arrested state.

---

## [Author Response]

Essential revisions:1) Sometimes these "refutation" papers don't get the most important point across. For this work, I think that the most important point is that the APH plays a crucial role in the translocation step. It further substantiates the point proposed in the earlier Alcock et al. 2013 paper that mutations in the APH causes an arrest at the assembled step and that the assembled state is stable even without the PMF, unless the substrate is transported.

We now emphasise this point in both the Abstract and the Discussion (last paragraph).

2) The authors did a good job in the Alcock et al. 2013 paper explaining why assembly can be seen in an uninduced system (i.e. no overexpressed substrate) if the transport step is impaired. That argument is important in the present manuscript and should at least be referred to in the Discussion. The authors don't want to leave the impression that TatA assembles without substrate. Triggering of assembly by substrate binding is generally accepted in the field.

We have now clarified this argument in both the Results (subsection “Direct observation of TatA assembly in charge zipper mutants”, end of third paragraph) and Discussion (last paragraph) sections.

3) Figure 2. The spheroplast fraction of several mutants (TatA K49D and TatA K52D) has a band that comigrates with processed CueO. Is this band an aborted transport product? This band is not present or very faint in samples from the most defective mutants (TatA EDD/KKK) so it doesn't seem to be a cross-reactive protein or a cytosolic CueO degradation product. The authors should comment on this band particularly if it provides some insight into the arrested state.

This additional band, which we saw occasionally in all of the strains, is most likely a degradation product of CueO. Through paying rigorous attention to experimental detail and working as rapidly as possible we are now routinely able to carry out export assays where we no longer see this band.

We have replaced the panels in Figure 2 with a new experiment where this band is absent.